# Effect of Butorphanol-Medetomidine and Butorphanol-Dexmedetomidine on Echocardiographic Parameters during Propofol Anaesthesia in Dogs

**DOI:** 10.3390/ani14091379

**Published:** 2024-05-03

**Authors:** Andrej Bočkay, Carlos Fernando Agudelo, Mária Figurová, Nela Vargová, Alexandra Trbolová

**Affiliations:** 1Small Animal Hospital, University of Veterinary Medicine and Pharmacy, Komenského 73, 041 81 Kosice, Slovakia; carlos.fernando.agudelo.ramirez@uvlf.sk (C.F.A.); maria.figurova@uvlf.sk (M.F.); nela.vargova@student.uvlf.sk (N.V.); alexandra.trbolova@uvlf.sk (A.T.); 2Small Animal Referral Centre Sibra, Na Vrátkach13, 841 01 Bratislava, Slovakia

**Keywords:** anaesthesia, butorphanol, dexmedetomidine, dog, echocardiography, medetomidine, propofol

## Abstract

**Simple Summary:**

Medetomidine and dexmedetomidine, commonly used in veterinary practice, induce sedation, muscle relaxation and analgesia. Reduced heart rate is one of the most common side effects of these anaesthetics. Other undesirable effects include increased vascular resistance, deterioration of blood supply to some tissues, and a decrease in cardiac output. These effects can be fatal in a critically ill patient. This study aimed to compare the haemodynamically depressive effects of butorphanol-medetomidine and butorphanol-dexmedetomidine combinations on echocardiographic parameters during propofol anaesthesia in dogs and to determine which anaesthetic produces fewer adverse cardiovascular effect.

**Abstract:**

This study compared the effects of butorphanol-medetomidine and butorphanol-dexmedetomidine combinations on echocardiographic parameters during propofol anaesthesia in dogs. The dogs were randomly divided into two groups. In the butorphanol-medetomidine (BM) group, butorphanol (0.2 mg/kg) and medetomidine (15 μg/kg) were intravenously administered; in the butorphanol-dexmedetomidine (BD) group, butorphanol (0.2 mg/kg) and dexmedetomidine (7.5 μg/kg) was used. Anaesthesia was induced with propofol and maintained with a constant-rate infusion of propofol (0.2 mg/kg/min). The echocardiographic parameters were assessed in conscious dogs (T_0_). Echocardiography was conducted again at 10 min post premedication (T_1_), followed by assessments at 30 (T_2_), 60 (T_3_), and 90 (T_4_) mins. The dogs were subjected to diagnostic procedures (radiography, computed tomography) under anaesthesia. A significant reduction in heart rate and cardiac output was noted in both groups at T_1_. There was no significant difference in the stroke volume between the BM and BD groups. The application of butorphanol-dexmedetomidine caused a significant increase in the left ventricular internal diameter in diastole and the diameter of the left atrium compared to that caused by butorphanol-medetomidine. This study documented that butorphanol-medetomidine and butorphanol-dexmedetomidine combinations caused similar reductions in heart rate and cardiac output in both groups. ‘New´ valvular regurgitation occurred following their administration.

## 1. Introduction

Medetomidine is a highly specific and potent α-2 adrenergic receptor agonist. Its α-2/α-1 selectivity ratio is 1620:1, and it is 5–10 times more selective than detomidine, clonidine, xylazine, or other anaesthetics with an α-2 adrenergic activity [1]. Like other α2-agonists, medetomidine provides reliable and profound sedation with a significant drug sparing effect on anaesthetic induction and maintenance agents [2]. The decrease in stroke index, cardiac index, total peripheral resistanceandtotal peripheral resistanceindex after medetomidine administration is dose-dependent [3].Medetomidine is usually used as a pre-anaesthetic medication before ketamine, sodium thiopental, propofol, or inhalation anaesthesia [4]. Medetomidine is a racemic mixture of dexmedetomidine and levomedetomidine. The right-handed optical isomer dexmedetomidine is pharmacologically active. The left-handed enantiomer levomedetomidineis considered to be inactive [5]. Dexmedetomidine was developed to reduce the side effects of medetomidine. In fact, even though the left-handed enantiomer is not pharmacologically active, it influences the pharmacokinetics and pharmacodynamics of dexmedetomidine. The absence of the levomedetomidine reduces the hepatic metabolic load and side effects [6]. In dogs, dexmedetomidine causes miosis following direct inhibition of parasympathetic stimulation of the iris and reduces intraocular pressure by the activation of the α2 receptors in the iris [7]. The activation of α-2 adrenergic receptors causes analgesia, anxiolysis, bradycardia, sedation and vasoconstriction [8,9,10]. Dexmedetomidine is very similar to medetomidine in terms of pharmacokinetic properties in dogs [11]. Half a dose of dexmedetomidine causes the same effects as the racemic mixture of medetomidine; therefore, it is considered twice as potent. High doses of levomedetomidine yield only a minimal analgesic and sedative effect [12,13]. Furthermore, contrasting results comparing the effects of these two substances have been documented in the literature [14,15].

The most serious cardiovascular effects of α-2 agonists are bradycardia, bradyarrhytmias (1st and 2nd degree of atrioventricular block) and decreased cardiac output (CO) [16]. Dexmedetomidine and medetomidine have similar haemodynamic effects. These effects include bradycardia, biphasic blood pressure response, increased systemic vascular resistance, and increased central venous pressure. The reported changes in the pulmonary arterial pressure or pulmonary capillary wedge pressure were minimal [17]. Another negative haemodynamic effect is a decrease in the perfusion of specific organs. Medetomidine has a reduced perfusion index in the abdominal aorta, renal arteries, cranial mesenteric and celiac arteries in dogs [18]. Dexmedetomidine reduces blood flow through the arteriovenous anastomoses and decreases brain, kidney, and skin perfusion [19].

Butorphanol is a synthetic opioid that acts mainly as a κ-opioid receptor agonist and a partial antagonist at the μ-opioid receptor [20]. Butorphanol is rapidly absorbed after intramuscular administration andit produces mild-to-moderate sedation with an onset time of <15 min [21], and its analgesic duration ranges from 1 to 3 h [22].A few studies using intramuscular and intravenous butorphanol in dogs have reported hypotension [23], but also cardiovascular stability [24]. Combined with medetomidine, butorphanol enhances the sedative effect of medetomidine [25] while preventing vomiting induced by medetomidine [26].

Propofol is a widely used anaesthetic for induction and total intravenous anaesthesia in dogs [27,28]. Propofol can be also used for stabilization of small animal neurological emergencies, including status epilepticus refractory to diazepam and phenobarbital [29]. Undesirable effects of propofol administration include reduction of systemic arterial pressure and respiratory depression [30]. A reduction in systemic arterial pressure is the result of decreased myocardial contractility, systemic vascular resistance, or sympathetic activity [31]. Propofol reportedly inhibits L-type Ca^2+^ channels [32], sarcoplasmic reticulum Ca^2+^ handling [33], and K^+^ channels [34] and increases Ca^2+^ sensitivity [35]. The positive and negative inotropic actions of propofol causes changes in the myocardium contractility [36,37,38].

The undesirable effects of medetomidine and dexmedetomidine can be fatal when administered to critically ill patients. Hence, thisstudy aimed to demonstrate which of the investigated anaesthetics cause less pronounced cardio-depressant effects and thus would be safer. The findings of this study may aid in mitigating the risk of anaesthesia associated with the use of medetomidine or dexmedetomidine, even during elective procedures. We presumed a comparable effect of the compared anaesthetics because of their similar pharmacokinetics.

## 2. Materials and Methods

This blind, prospective, randomised study was approved by the Ethics Committee of University of Veterinary Medicine and Pharmacy in Košice (EKVP/2023-20). The study followed the guidelines of the State Veterinary and Food Administration of the Slovak Republic, and the instructions and international directives of the European Parliament and the Council of the EU. Written informed consent authorising study participation was obtained from each dog’s owner.

Forty client-owned dogs of breed Tatra hound were included in this study. All dogs included in the study were subjected to echocardiography and electrocardiography (ECG) examinations performed by an experienced veterinarian. The age and weight of the dogs were recorded. Dogs with initial ECG abnormalities, echocardiographic pathology, or those that could not be handled were excluded. Dogs were randomly allocated into two groups: butorphanol-medetomidine (BM) and butorphanol-dexmedetomidine (BD), using an online randomization tool (https://www.graphpad.com/quickcalcs/randomize1/, accessed on 19 May 2023) prior to initiating the study.

Dogs were admitted to the University Veterinary Teaching Hospital on the morning of the diagnostic procedures (radiography, computed tomography). The diagnostics were conducted to standardise the characteristics of the Tatra hound breed. Food was withheld by the owners for 12 h prior to anaesthesia. Water was withheld 2 h before premedication. All dogs included in the study underwent initial echocardiographic (Esaote MyLabX5vet, Genova, Italy) and ECG (SEIVA EKG Praktik Veterinary USB, Prague, Czech Republic) examinations in a fully conscious state performed by an experienced veterinarian before the administration of premedication (T_0_). Two intravenous catheters were placed in the cephalic veins following aseptic preparation prior to premedication. Dogs in the BM group were premedicated with butorphanol 0.2 mg/kg (Butomidor; Richter Pharma AG, Wels, Austria) plus medetomidine 15 μg/kg (Cepetor; CP-Pharma Handelsges, Germany) and dogs in the BD group were premedicated with butorphanol 0.2 mg/kg (Butomidor; Richter Pharma AG, Wels, Austria) plus dexmedetomidine 7.5 μg/kg (Dexdomitor; Orion Corporation, Espoo, Finland) intravenously. Ten minutes after the premedication, a second echocardiographic assessment (T_1_) was performed. Anaesthesia was induced with propofol (Propofol; Fresenius Kabi GmbH, Graz, Austria) administered intravenously until endotracheal intubation was achieved. General anaesthesia was maintained with a constant-rate infusion of propofol 0.2 mg/kg/min using syringe driver (Syringe pump SN-50 F6, Hamburg, Germany). All dogs were supported on 100% oxygen. Lactated Ringer’s solution (Compound Sodium Lactate Ringer-Lactat; B. Braun Melsungen AG, Melsungen, Germany) was initiated at aconstant rate of 5 mL/kg/h. Anaesthetic monitoring (Mindray BeneView T8, Hamburg, Germany) consisted of ECG (3-lead placement), respiratory rate, end-tidal carbon dioxide (side-stream capnography) and haemoglobin oxygen saturation (probe placed on the tongue). These parameters were recorded every 5 min during the procedure. Further echocardiographic examinations were performed 30 (T_2_), 60 (T_3_) and 90 (T_4_) mins after T_1_ assessment.

### Echocardiographic Examination

All echocardiographic assessments were performed in left and right lateral recumbency in a quiet, dark room and included 2D, M-mode, and Doppler images from the right parasternal long and short axes and the left apical view [39]. Measurements of the left ventricle, interventricular septum at end-systole, and left ventricular posterior wall thickness at end-systole and end-diastole were obtained using M-mode at the level of the papillary muscles from the right parasternal short-axis image. The left atrial, aorta diameter, and right ventricular outflow tract dimensions were obtained from the short axis view at the heart base. Simpson’s method was used for disc-derived left ventricular end-diastolic volume and left ventricular end-systolic volume, also the left ventricular outflow tract were calculated using the right parasternal long-axis view [40]. Stroke volume and cardiac output were estimated with the help of ultrasound software. Pulsed-wave Doppler was used to evaluate the blood flow velocities in the pulmonary artery from the right parasternal short-axis at the level of the heart base, the mitral and tricuspid valves from the left apical views, and the aortic blood flow velocities by using continuous-wave Doppler. All the cardiac valves were evaluated using colour-flow Doppler in the caseof insufficiency. For all echocardiographic measurements, a lead II electrocardiogram was recorded simultaneously.

Data were analysed using GraphPad Prism 8 (GraphPad Software, La Jolla, CA, USA, ver. 8.0.1.244). Numerical data are expressed as medians with ranges. Data were tested for normality using theShapiro–Wilk test. To compare all echocardiographic variables at the same time points (T_0_, T_1_, T_2_, T_3_, and T_4_) between the BM and BD groups, the Mann–Whitney U test was used. A Kruskal–Wallis test followed by post hoc Dunn’s test with Bonferroni correction was used to compare the values for each variable over time (T_0_, T_1_, T_2_, T_3_, and T_4_) within each group. The incidence of valvular regurgitation between groups was compared using Fisher’s exact test. The level of significance was set at *p* < 0.05.

## 3. Results

Thirty-seven dogs out of forty completed the study. Two dogs in the BM group and one dog in BD group were excluded owing to congenital, albeit irrelevant, heart disease. The age of dogs varied from 12–72 months with a median of 12 months in the BM and from 12–84 months with a median of 24 months in the BD group. The median body weights of the BM and BD group dogs were 15.48 kg (12–28.5 kg) and 16.25 kg (11.2–20.8 kg), respectively. There were seven (39%) males and 11 (61%) females in the BM group and eight males (42%) and 11 (58%) females in the BD group. All dogs were intact.

In both groups, there was a significant reduction in HR and CO at T_1_ compared to those at T_0_, and this reduction persisted to the last assessment at T_4_. HR, CO, and SV are summarised in Table 1. LVIDd was larger in the BD group than in the BM group at T_1_ and T_4_. The left atrial diameter was greater in the BD group at T_1_ and T_4_. The LVOT Vmax and LVOT max PG were higher at baseline than at subsequent time points in the BM and BD groups. A reduction in the MV E Vel was observed within both groups at T_2_ and T_3_ compared to that at T_0_; however, it was no longer present at T_4_. A decrease in the MV A Vel was observed at T_2_ in both groups. In the BM group, the TV E Vel decreased significantly at T_2_ and T_3_. Echocardiographic measurements are presented in Table 2, Table 3 and Table 4.

Valve regurgitation was absent initially; however, it developed in some dogs following premedication. No significant difference was found in the incidence of valvular regurgitation between the BM and the BD groups. In the BM group, MR was found in 33% of the dogs at T_1_ and T_2_, 22% of the dogs showed regurgitation at T_3_, and 11% of the dogs showed MR at T_4_. The degree of regurgitation was set at ¼. In the BM group, TR was diagnosed in 6% of dogs at T_1_, 22% of dogs at T_2_, 28% at T_3_, and 11% at T_4_; the degree of regurgitation was set at ¼. In the BD group, MR was found in 50% of dogs at T_1_, 22% ofdogs at T_2_; 28% of dogs showed regurgitation at T_3_ and 22% of dogs at T_4_. The degree of regurgitation was set at ¼. In the BD group, TR was diagnosed in 33% of dogs at T_1_, 22% of dogs at T_2_, 17% at T_3_, and 6% at T_4_. The degree of the TR was set to ¼. In the group BD, aortic regurgitation was diagnosed in 6% of dogs at T_2_. At this time point, pulmonic regurgitation was diagnosed in 6% of dogs in the BD group. A summary of the incidences of valvular regurgitation is presented in Table 5.

## 4. Discussion

The findings of this study indicated that butorphanol-medetomidine and butorphanol-dexmedetomidine caused very similar changes in echocardiographic parameters.

Application of alpha-2- agonist causes depression of the cardiovascular system. The negative effects of this group of anaesthetics include bradycardia, bradyarrhythmia, increased vascular resistance, decreased perfusion of specific organs, and decreased CO [10,11,12]. A significant reduction in the HR was observed in both groups after premedication which persisted until the last examination. This depressant cardiovascular effect of medetomidine and dexmedetomidine has been documented previously in several studies [13,41,42,43]. CO is reduced after premedication, and this reduction may be associated with a decrease in HR and an increase in vascular resistance, but not with a direct depression of myocardial contractility [44]. Thus, CO may have decreased in our study because of a positive association with a decrease in HR, as observed in several studies [45,46,47,48].

An increase in theLVIDd dimension and LA diameter was observed in both treated groups following premedication at T_1_ with a more pronounced effect observed withthe butorphanol-dexmedetomidine combination. Our findings correlated with the increase in the left atrial size and volume described in other studies [49,50,51]. For instance, an increase in the size of the left atrium was reported in a study using medetomidine and xylazine when evaluated via echocardiography [50]. Dexmedetomidine in the BD group resulted in a prolonged filling time of the right ventricle, suggesting a potential enhancement in diastolic function. A lower HR, as observed in our study, may be associated with an increased diastolic filling period, which was reported as a possible cause of cardiac enlargement in a study investigating the effects of dexmedetomidine [49]. A decreased HR improves the chance of the heart achieving greater ventricular filling. Another parameter associated with the effect on the diastolic function of the left ventricle was atrial contraction (MV A Vel), which was reduced in the BD group at T_3_. Both groups showed a reduction in atrial contractions after premedication. This could also indicatea negative inotropic effect on the atrial myocardium; however, this has not yet been reported.

In the present study, valvular regurgitation occurred in both the groups after the first administration. Appearance of ‘new’ valvular regurgitation has been reported in a study comparing two different doses of dexmedetomidine (5 μg/kg vs. 10 μg/kg) in combination with butorphanol. Similar to our study, no dogs in the mentioned study had echocardiographic evidence of regurgitation, but >50% in the low-dose group and >80% in the high-dose group showed regurgitation after application [52]. Furthermore, apparent mitral and pulmonic regurgitation have been also reported after intravenous injection of dexmedetomidine at a dosage of 250 μg/m^2^ (approximately 10 μg/kg) in all dogs included in the study. Additionally, tricuspid (5/6) and aortic regurgitation (4/6) were observed [49]. Tricuspid and mitral regurgitation have also been reported in one of ten dogs after asingle injection of dexmedetomidine (1–2 μg/kg) followed by a continuous rate infusion (1–2 μg/kg/h) in atwo-dimensional phase-contrast magnetic resonance imaging study [53]. In a previous study, an increase in preload and end-diastolic volume was reported as a possible cause of mitral regurgitation. Another cause may be the excessive accumulation of blood in the heart and associated enlargement of the ventricles. This enlargement can create a space between the valve leaflets and cause regurgitation [49]. Mitral regurgitation was also noted in two-thirds of dogs in a study evaluating the effect of xylazine and medetomidine on echocardiographic parameters in six healthy dogs [50]. Based on our results and those of other studies, we assume that the incidence and degree of regurgitation are common features when medetomidine and dexmedetomidine are used as premedication and may be associated with the dosage.

Negative effects on left heart contractility were also observed after the application of propofol, such as a reduction in aortic flow and atrial contraction, in both treated groups. This may have been caused by the direct negative inotropic effect of propofol on the left ventricle. However, this reduction may have been relatively small [54]. Based on these results, it can be concluded that propofol negatively affects both systolic and diastolic heart functions.

This study had several limitations. Firstly, there was extreme homogeneity among the groups of dogs. Only Tatra hounds were included in the study. The dogs were comparable in terms of age and weight. The results of our study may be applied to similar breeds of dogs, but haemodynamics differ among breeds. Secondly, the dogs in our study were subjected diagnostic procedures without nociceptive stimuli. Our study cannot guarantee that the effects induced by medetomidine and dexmedetomidine will be the same during surgery, when nociceptive stimuli can occur and alter the haemodynamics. Further studies are warranted in other dog breeds and dogs undergoing procedures with varying degrees of invasiveness.

## 5. Conclusions

This study demonstrated that the intravenous administration of butorphanol-medetomidine and butorphanol-dexmedetomidine combination caused similar reductions in HR, CO and incidence of ‘new’ valvular regurgitation during propofol-maintained general anaesthesia. The butorphanol-dexmedetomidine combination may induce greater enlargement of the left side of the heart (LVIDd and LA Diam) than does butorphanol-medetomidine. Propofol may also negatively affect systolic and diastolic heart functions.

## Figures and Tables

**Table 1 animals-14-01379-t001:** The effects of butorphanol (0.2 mg/kg IV)-medetomidine (15 μg/kg IV) and butorphanol (0.2 mg/kg IV)-dexmedetomidine (15 μg/kg IV) combinationsonheart rate, stroke volume, and cardiac output. First echocardiographic examination was carried out in fully conscious dogs in T_0_. Second examination was carried out10 min after administration of premedication in T_1_. After the second echocardiographic examination, general anaesthesia was induced (dose-to-effect of endotracheal intubation) and maintained by propofol CRI (0.2 mg/kg/min). Following echocardiographic assessments were performed at 30 (T_2_), 60 (T_3_) and 90 (T_4_) mins after premedication.

	T_0_		T_1_		T_2_		T_3_		T_4_	
	*BM*	*BD*	*BM*	*BD*	*BM*	*BD*	*BM*	*BD*	*BM*	*BD*
HR (bpm)	110 (68–135)	107 (54–144)	43 ^(a)^ (30–95)	43 ^(a)^ (25–90)	53 ^(a)^ (20–85)	42 ^(a)^ (26–80)	61 ^(a)^ (26–80)	55 ^(a)^ (31–81)	53 ^(a)^ (34–83)	52 ^(a)^ (30–74)
SV (ml)	21.3 (9.1–33.7)	23.2 (12.3–42)	20.5 (6.6–34.2)	25.6 (7.1–38.4)	18.8 (6.3–39.8)	23.5 (8.1–33.6)	21.9 (9.2–36.6)	22.1 (10.6–43.6)	17.3 (12.2–37.5)	24.1 (16.2–42)
CO (l/min)	2.1 (0.8–3.2)	2.1 (0.9–3.7)	1.0 ^(a)^ (0.4–2.1)	1.1 ^(a)^ (0.3–2.4)	1.1 ^(a)^ (0.3–2.5)	0.9 ^(a)^ (0.3–2.2)	1.0 ^(a)^ (0.4–2.7)	1.2 ^(a)^ (0.3–2.7)	0.9 ^(a)^ (0.4–2.5)	1.0 ^(a)^ (0.6–3.0)

Abbreviation: ^(a)^: significant difference from baseline (T_0_), BD: butorphanol + dexmedetomidine group, BM: butorphanol + medetomidine group, CO: cardiac output, HR: heart rate, SV: stroke volume, T_0_: baseline, T_1_: 10 min after premedication, T_2_: 30 min after premedication, T_3_: 60 min after premedication, T_4_: 90 min after premedication. Numerical data are presented as medians with ranges.

**Table 2 animals-14-01379-t002:** Changes of selected M-mode echocardiographic measurements after butorphanol (0.2 mg/kg IV)-medetomidine (15 μg/kg IV) and butorphanol (0.2 mg/kg IV)-dexmedetomidine (15 μg/kg IV) administration. First echocardiographic examination was carried outin fully conscious dogs in T_0_. Second examination was carried out10 min after administration of premedication in T_1_. After the second echocardiographic examination, general anaesthesia was induced (dose-to-effect of endotracheal intubation) and maintained by propofol CRI (0.2 mg/kg/min). Following echocardiographic assessments were performed 30 (T_2_), 60 (T_3_) and 90 (T_4_) mins after premedication.

	T_0_		T_1_		T_2_		T_3_		T_4_	
	*BM*	*BD*	*BM*	*BD*	*BM*	*BD*	*BM*	*BD*	*BM*	*BD*
RVIDd (mm)	10.3 (3.4–15.5)	8.6 (4.4–18.1)	12.4 (5.5–22)	15.5 (4.5–19.4)	11.9 (3.2–20.7)	12.3 (2.9–23.3)	11 (2.8–23.2)	14.9 (3.2–24.6)	10.1 (4.5–17)	10 (3.2–19.4)
LVIDd (mm)	35.1 (31–42.6)	37.9 (29.9–42.6)	35.6 * (30.5–43.9)	38.8 * (31.1–49.1)	37.7 (31.7–43.3)	39.4 (31.5–45.2)	36 (25.2–45.2)	39.1 (16.2–45.2)	37.3 * (27.1–44.6)	40.4 * (33–45.9)
IVSs (mm)	12.4 (9.6–16.8)	13.6 (10.5–16.2)	11 (8.5–15.3)	11.6 (9–14.9)	11.2 (7.3–16.4)	12.1 (9–14.9)	10.3 (7.9–14.2)	11.3 (7.3–15.5)	11.8 (6.2–16.2)	12.3 (7.8–15.5)
LVPWs (mm)	12.6 (9.6–16.7)	11.6 (9–16.2)	10.3 (7.8–14.5)	11 (6.8–14.2)	11.7 (7.9–17.4)	12.3 (9.7–14.2)	11.5 (7.3–13.6)	12.3 (8.4–14.2)	11.3 (7.3–18.1)	12.3 (8.4–15.5)
IVSd (mm)	9 (7.3–13.1)	9.7 (7.7–12.3)	9.6 (6.3–17)	8.4 (6.8–1.3)	9 (6.5–12.4)	9 (5.8–13.6)	9.2 (7.8–15.5)	9.7 (6.1–13.6)	9.4 (7.3–11.6)	9 (5.8–12.3)
LVPWd (mm)	7.3 (5.1–10.9)	7.8 (5.8–10.3)	7.9 (5.7–12.9)	7.1 (5.8–9.7)	7.9 (5.1–13.8)	7.8 (6.3–9.7)	8.4 (4.8–10.7)	7.8 (5.8–12.3)	8.2 (5.1–11.3)	8.4 (6.5–10.3)
LVIDs (mm)	24.3 (18.1–29.1)	26.5 (22.6–31)	30.4 (18.7–37.5)	30.4 (26–38.1)	29.6 (18.7–37.5)	29.1 (24.2–40.1)	27.8 (25.2–37.5)	30.1 (23.3–39.4)	26.3 (20.4–32.2)	27.8 (21.8–38.1)

Abbreviation: BD: butorphanol + dexmedetomidine group, BM: butorphanol + medetomidine group, IVSd: interventricular septum in diastole, IVSs: interventricular septum in systole, LVIDd: left ventricular internal diameter in diastole, LVPWd: left ventricular posterior wall in diastole, LVPWs: left ventricular posterior wall in systole, RVIDd: right ventricular internal diameter in diastole, T_0_: baseline, T_1_: 10 min after premedication, T_2_: 30 min after premedication, T_3_: 60 min after premedication, T_4_: 90 min after premedication, and *: significant difference between groups. Numerical data are presented as medians with ranges.

**Table 3 animals-14-01379-t003:** Changes of selected Doppler echocardiographic measurementsafter butorphanol (0.2 mg/kg IV)-medetomidine (15 μg/kg IV) and butorphanol (0.2 mg/kg IV)-dexmedetomidine (15 μg/kg IV) administration. First echocardiographic examination was carried outin fully conscious dogs in T_0_. Second examination was carried out10 min after administration of premedication in T_1_. After the second echocardiographic examination, general anaesthesia was induced (dose-to-effect of endotracheal intubation) and maintained by propofol CRI (0.2 mg/kg/min). Following echocardiographic assessments were performed 30 (T_2_), 60 (T_3_) and 90 (T_4_) mins after premedication.

	T_0_		T_1_		T_2_		T_3_		T_4_	
	*BM*	*BD*	*BM*	*BD*	*BM*	*BD*	*BM*	*BD*	*BM*	*BD*
LVOT Vmax (m/s)	0.94 (0.67–1.16)	0.85 (0.59–1.25)	0.64 ^(a)^ (0.28–1.19)	0.57 ^(a)^ (0.4–1.08)	0.42 ^(a)^ (0.29–0.88)	0.57 ^(a)^ (0.29–0.95)	0.5 ^(a)^ (0.1–0.95)	0.55 ^(a)^ (0.34–0.91)	0.55 ^(a)^ (0.4–0.83)	0.65 ^(a)^ (0.44–0.83)
LVOT Vmax PG (mmHg)	3.5 (1.8–5.4)	2.9 (1.4–6.2)	1.6 ^(a)^ (0.4–5.7)	1.3 ^(a)^ (0.6–4.7)	0.8 ^(a)^ (0.3–3.1)	1.3 ^(a)^ (0.3–3.6)	1 ^(a)^ (0.5–3.6)	1.2 ^(a)^ (0.5–3.3)	1.15 ^(a)^ (0.6–2.8)	1.7 ^(a)^ (0.8–2.8)
MV E Vel (m/s)	0.65 (0.45–0.8)	0.66 (0.46–0.86)	0.5 (0.23–0.76)	0.58 (0.4–0.72)	0.47 ^(a)^ (0.29–0.74)	0.46 ^(a)^ (0.25–0.8)	0.45 ^(a)^ (0.24–0.59)	0.45 ^(a)^ (0.22–0.75)	0.54 (0.39–0.77)	0.54 (0.37–0.91)
MV A Vel (m/s)	0.44 (0.28–0.64)	0.44 (0.24–0.77)	0.3 ^(a)^ (0.16–0.53)	0.27 ^(a)^ (0.2–0.51)	0.29 (0.21–0.54)	0.29 (0.21–0.45)	0.29 * (0.2–0.52)	0.15 * (0.13–0.34)	0.25 (0.15–0.49)	0.24 (0.17–0.46)
MV E/A	1.42 (1.06–2.14)	1.41 (1–3)	1.74 (1–3.08)	1.74 (0.95–3.25)	1.59 (0.84–3.24)	1.78 (0.67–2.78)	1.65 (0.84–2.27)	1.88 (1.06–3)	1.78 (1.14–2.94)	1.95 (1.05–3.1)
MV Dec Time (ms)	100 (44–148)	124 (76–184)	106 (58–180)	92 (72–172)	114 (52–152)	106 (76–152)	84 (38–204)	96 (64–224)	91 (44–192)	106 (72–197)
TV E Vel (m/s)	0.5 (0.37–0.75)	0.48 (0.35–0.67)	0.36 * (0.19–0.61)	0.42 * (0.29–0.74)	0.39 * (0.27–0.58)	0.38 * (0.23–0.57)	0.37 (0.24–0.55)	0.39 (0.21–0.55)	0.45 (0.25–0.6)	0.36 (0.25–0.56)
TV E/A	1.4 (0.69–2.75)	1.7 (1.26–2.35)	1.56 (0.83–4.83)	2.15 (0.96–2.76)	1.32 (0.72–2.17)	1.39 (0.89–3)	1.26 (0.68–2.08)	1.54 (0.85–2.59)	1.9 (1.09–2.67)	1.6 (0.74–2.79)
TV E Dec Time (ms)	118 (52–184)	100 (72–148)	114 (60–188)	144 (84–192)	128 (66–236)	136 (72–272)	120 (64–216)	172 (60–256)	144 (104–232)	156 (72–232)

Abbreviation: ^(a)^: significant difference from baseline (T_0_), BD: butorphanol + dexmedetomidine group, BM: butorphanol + medetomidine group, LVOT Vmax PG: left ventricular outflow tract peak gradient, LVOT Vmax: left ventricular outflow tract maximum velocity, MV A Vel: mitral valve A velocity, MV Dec Time: mitral valve deceleration time, MV E Vel: mitral valve E velocity (early diastolic flow peak velocity of the mitral valve), MV E/A: mitral valve E-wave/A-wave ratio, T_0_: baseline, T_1_: 10 min after premedication, T_2_: 30 min after premedication, T_3_: 60 min after premedication, T_4_: 90 min after premedication, TV E Dec Time: tricuspidal valve deceleration time, TV E Vel: tricuspidal valve E velocity, TV E/A: tricuspid valve E-wave/A-wave ratio, and *: significant difference between groups. Numerical data are presented as medians with ranges.

**Table 4 animals-14-01379-t004:** Changes of selected B mode echocardiographic measurements after butorphanol (0.2 mg/kg IV)-medetomidine (15 μg/kg IV) and butorphanol (0.2 mg/kg IV)-dexmedetomidine (15 μg/kg IV) administration. First echocardiographic examination was carried outin fully conscious dogs in T_0_. Second examination was carried out10 min after administration of premedication in T_1_. After the second echocardiographic examination, general anaesthesia was induced (dose-to-effect of endotracheal intubation) and maintained by propofol CRI (0.2 mg/kg/min). Following echocardiographic assessments were performed 30 (T_2_), 60 (T_3_) and 90 (T_4_) mins after premedication.

	T_0_		T_1_		T_2_		T_3_		T_4_	
	*BM*	*BD*	*BM*	*BD*	*BM*	*BD*	*BM*	*BD*	*BM*	*BD*
RVOT Diam (mm)	19 (12.1–25.5)	19.3 (16–23.4)	19.5 (10.2–25.8)	20.2 (17.4–25.9)	19 (10.9–22.8)	21 (16.8–23.6)	19.4 (12–23.3)	20.9 (16.7–22.5)	19.1 (13.1–23.7)	20.5 (17.4–23.4)
LVOT Diam (mm)	17.1 (12.5–20.6)	18.2 (14.1–21.7)	17.4 (10.9–21.9)	18.4 (14.2–21.2)	16.3 (12.5–22.5)	18.2 (15.3–20.6)	16.9 (13–19.7)	18.2 (15.8–21.8)	16.4 (12.7–20.3)	17.3 (15–20.5)
Ao Diam (mm)	20.15 (16.2–24.4)	20.45 (17.2–24.2)	19.6 (13.7–25)	21.6 (17.3–25.2)	20.2 (12.2–24.1)	21.4 (17.6–25.4)	20.2 (15.1–25.4)	21.4 (17.9–23.3)	19.3 (14.3–22.6)	21 (18.3–24.5)
LA Diam (mm)	23.7 (19–33.2)	25.6 (21.9–34.5)	28.3 * (19.7–34.6)	29.4 * (19.8–36)	27.9 (20.6–34.4)	29.5 (19.8–26.8)	27.1 (20.3–34.5)	29.6 (22.9–32.3)	24.9 * (19.1–31)	28.6 * (18.9–34.9)
LA/Ao	1.22 (1.07–1.62)	1.32 (1.02–1.65)	1.45 (1.09–1.88)	1.42 (1.09–1.61)	1.41 (1.06–1.98)	1.39 (1.1–1.58)	1.34 (1.19–1.61)	1.36 (1.25–1.53)	1.33 (1.02–1.51)	1.41 (1.02–1.61)

Abbreviation: AO Diam: aortic diameter, BD: butorphanol + dexmedetomidine group, BM: butorphanol + medetomidine group, LA Diam: left atrium diameter, LA/Ao: left atrial-to-aortic root diameter ratio, LVOT Diam: left ventricular outflow tract diameter, RVOT Diam: right ventricular outflow tract diameter, T_0_: baseline, T_1_: 10 min afterpremedication, T_2_: 30 min after premedication, T_3_: 60 min after premedication, T_4_: 90 min after premedication, and *: significant difference between groups. Numerical data are presented as medians with ranges.

**Table 5 animals-14-01379-t005:** Onset of valvular regurgitation after butorphanol (0.2 mg/kg IV)-medetomidine (15 μg/kg IV) and butorphanol (0.2 mg/kg IV)-dexmedetomidine (15 μg/kg IV) administration. First echocardiographic examination was carried outin fully conscious dogs in T_0_. Second examination was carried out10 min after administration of premedication in T_1_. After the second echocardiographic examination, general anaesthesia was induced (dose-to-effect of endotracheal intubation) and maintained by propofol CRI (0.2 mg/kg/min). Following echocardiographic assessments were performed 30 (T_2_), 60 (T_3_) and 90 (T_4_) mins after premedication.

	T_0_		T_1_		T_2_		T_3_		T_4_	
	*BM*	*BD*	*BM*	*BD*	*BM*	*BD*	*BM*	*BD*	*BM*	*BD*
MR	0	0	6 (33%)	9 (50%)	6 (33%)	4 (22%)	4 (22%)	5 (28%)	2 (11%)	4 (22%)
TR	0	0	1 (6%)	6 (33%)	4 (22%)	4 (22%)	5 (28%)	3 (17%)	2 (11%)	1 (6%)
AR	0	0	0	0	0	1 (6%)	0	0	0	0
PR	0	0	0	0	0	1 (6%)	0	0	0	0

Abbreviation: AR: aortic regurgitation, BD: butorphanol (0.2 mg/kg) + dexmedetomidine (7.5 μg/kg) group, BM: butorphanol (0.2 mg/kg) + medetomidine (15 μg/kg) group, MR: mitral regurgitation, PR: pulmonic regurgitation, T_0_: baseline, T_1_: 10 min after premedication, T_2_: 30 min after premedication, T_3_: 60 min after premedication, T_4_: 90 min after premedication, and TR: tricuspid regurgitation. The numbers represent the quantity and percentage of dogs with regurgitation.

## Data Availability

Data are available from the corresponding author upon reasonable request.

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
