# Peer review of "Effect of Butorphanol-Medetomidine and Butorphanol-Dexmedetomidine on Echocardiographic Parameters during Propofol Anaesthesia in Dogs"

_animals, 2024, doi:10.3390/ani14091379_

Round 1
Reviewer 1 Report
Comments and Suggestions for Authors
Introduction section
Line 61 - The application of propofol can reduce systemic arterial pressure and 61 respiratory depression [16]. What are the authors trying to say by this?
Lines 71-73 - The justification presented to conduct the study is too simplistic. A quick review over pubmed found a little less than 100 studies related to the use of medetomidine in dogs, with interest in cardiovascular parameters. Furthermore, the cardiovascular changes related to the use of alfa-2 agonists are well known to the veterinary medicine, despite the drug or the species studied.
Table titles are incomplete.
Where are the statistical differences presented in the tables. Are the authors assuming statistical differences in comparison to T0? How were the animals prior to the study? Were they familiar to the environment? Were there an adaptation period prior to measurements?
Discussion section
Line 233 - The induction of general anaesthesia and maintenance with propofol did not contribute to a greater decrease HR. Based on what did the authors make this assumption?
Lines 234-235 - This depressant cardiovascular effect of medetomidine and dexmedetomidine has been documented previously in several studies [27-30]. So what's the novelty here?
DIscussion is merely results confrontation between previous studies.
Conclusion section
Wouldn't call evidence of NEW regurgitation, since this feature had been previously reported as presented by the authors.
There is no control group. The effects of propofol over cardiovascular function could be solely assessed in order to present changes associated with the drug.
Transient cardiovascular changes attributed to the use of alpha-2 agonists have been studied to greater extent. For how long have the changes been observed? There's no mention of this. Statistical results have not been presented in the tables, so is quite difficult to observe and identify statistical differences. Furthermore, there has been comparisons between data from awaken dogs and following sedation/anesthesia. What's the level of stress in these dogs during T0 echocardiography? Where theses dogs conditioned to echocardiographic examinations? Would that not be a study limitation?
Reviewer 2 Report
Comments and Suggestions for Authors
I read the paper very carefully.
I have to say that in my opinion it needs some additional work to make it publishable. In particular, the whole paper is structured around very old references that are now outdated or at least much debated. This poses some questionable claims, or at the very least they need to be argued by taking into consideration all the valid references in the literature. Statistics is another point to be implemented. It is not very clear, which exposes the results to doubts as to their accuracy. Surely they are done with criterion, it is sufficient to present the statistics more clearly. Other minor comments are given in the file. I hope the revision will help implement the quality of the paper. I think a good result can be achieved with little effort.
Good work
Reviewer 3 Report
Comments and Suggestions for Authors
The article is interesting and contributing. The effect of selective alpha-2 adrenergic agonists has been studied for a long time, but some of their properties have not yet been described in detail.
Title
In my opinion, the title does not accurately describe the content of the study. Butorphanol should also be mentioned in the title. I recommend changing the title to "Effects of medetomidine-butorphanol and dexmedetomidine-butorphanol on echocardiographic parameters during propofol anaesthesia in dogs".
Abstract
L 17, 28, 29, 30 – I recommend adding a butorphanol
L 26, 27 – How did you non-invasively measure cardiac output or stroke volume? I can find no mention in the text.
Keywords
Add butorphanol
Introduction
Complete the paragraph about the effects of butorphanol.
L 45-46 – I recommend removing the sentence " However, the clinical benefits of medetomidine or dexmedetomidine are unclear.”
L 69 – I recommend removing “or those with cardiovascular diseases”. In some diseases (HCM) alpha-2 agonists can be used.
Material and Methods
L 77 – I recommend removing “single”.
L 83 – Why did you use 40 dogs? Please complete the sample size collection.
L 93 – What does “Non-invasive diagnostics” mean?
L 100, 102 – Complete the abbreviations (BM, BD) for individual groups.
L 105 – What was the total dose of propofol for induction of anesthesia? Did it differ between groups? It could have a major impact on the results.
L 106 – “with a constant rate infusion” – How was propofol administered? Using a syringe driver? Type?
L 109 – Why did you infuse at 10 ml/kg/h. For healthy dogs, 5 ml/kg/h is recommended. Could this affect the results?
L 110-111 – What type of vital function monitor did you use and how were the individual parameters measured? – please, describe more precisely.
L 133-134 – Why do you give “median with ranges” once and “median and IQR” the second time. I recommend stating only the median (ranges).
Tables
The legend of each table must contain complete information so that it is understandable even if presented separately. E.g., HR, SV, CO after premedication with medetomidine (dose) with butorphanol (dose) or dexmedetomidine (dose) with butorphanol (dose), induction with propofol (dose) and maintenance using constant rate infusion of propofol (dose) in dogs.
All the legends are also missing what the given numbers mean, please complete.
The significance of the differences is not indicated in the tables, please complete.
Conclusion
L 293-294 – Complete butorphanol and propofol.
References
For references, usually only the three authors are mentioned, for the rest "et al."
L 393-395 – Replace uppercase letters with lowercase letters in the title.
The manuscript is beneficial for the professional community as well as for practice. Due to the missing information in the Material and methods and in the Tables, however, I recommend revising, correcting and supplementing it. Therefore I recommend another review after rewriting and revisions.
Round 2
Reviewer 2 Report
Comments and Suggestions for Authors
Dear authors,
I red the new version of paper.
I have seen you effors to improve the papers.
Well done
Reviewer 3 Report
Comments and Suggestions for Authors
Dear authors, thank you for the careful corrections that significantly improve the manuscript. I have no longer any comments on the corrected manuscript.